# Improving teaching about medically unexplained symptoms for newly qualified doctors in the UK: findings from a questionnaire survey and expert workshop

Katherine Yon,[1] Stephanie Habermann,[1] Joe Rosenthal,[1] Kate R Walters,[1] Sarah Nettleton,[2] Alex Warner,[1] Kethakie Lamahewa,[1] Marta Buszewicz[1]

► Prepublication history and additional material is available. To view please visit the journal (http://dx.doi.org/10.1136/bmjopen-2016-014720).

[1]Research Department of Primary Care and Population Health, UCL, London, UK
[2]Department of Sociology, University of York, York, UK

**Correspondence to**
Katherine Yon;
katherine.yon.12@ucl.ac.uk

## ABSTRACT

**Objectives** Medically unexplained symptoms (MUS) present frequently in healthcare, can be complex and frustrating for clinicians and patients and are often associated with overinvestigation and significant costs. Doctors need to be aware of appropriate management strategies for such patients early in their training. A previous qualitative study with foundation year doctors (junior doctors in their first 2 years postqualification) indicated significant lack of knowledge about this topic and appropriate management strategies. This study reviewed whether, and in what format, UK foundation training programmes for newly qualified doctors include any teaching about MUS and sought recommendations for further development of such training.

**Design** Mixed-methods design comprising a web-based questionnaire survey and an expert consultation workshop.

**Setting** Nineteen foundation schools in England, Wales and Northern Ireland

**Participants** Questionnaire administered via email to 155 foundation training programme directors (FTPDs) attached to the 19 foundation schools, followed by an expert consultation workshop attended by 13 medical educationalists, FTPDs and junior doctors.

**Results** The 53/155 (34.2%) FTPDs responding to the questionnaire represented 15 of the 19 foundation schools, but only 6/53 (11%) reported any current formal teaching about MUS within their programmes. However, most recognised the importance of providing such teaching, suggesting 2–3 hours per year. All those attending the expert consultation workshop recommended case-based discussions, role-play and the use of videos to illustrate positive and negative examples of doctor–patient interactions as educational methods of choice. Educational sessions should cover the skills needed to provide appropriate explanations for patients' symptoms as well as avoid unnecessary investigations, and providing information about suitable treatment options.

**Conclusions** There is an urgent need to improve foundation level training about MUS, as current provision is very limited. An interactive approach covering a range of topics is recommended, but must be delivered within a realistic time frame for the curriculum.

## Strengths and limitations of this study

► To our knowledge, this is the first study to examine the provision of early postgraduate teaching on the topic of medically unexplained symptoms for newly qualified doctors.
► This study highlights the lack of training currently taking place on this topic for UK foundation year doctors and draws attention to the need to include the topic in the national curriculum. This is likely to be of relevance in other countries beyond the UK.
► Only around a third of the programme directors approached responded to the survey, although these respondents represented 15 of the 19 foundation schools across England, Wales and Northern Ireland.
► Our linked studies have produced recommendations for the content and format of an educational intervention for newly qualified doctors, but this needs to be further developed and formally evaluated to determine its impact.

## INTRODUCTION

UK medical foundation schools provide 2 years of compulsory postgraduate training for newly qualified junior doctors. Training is delivered in accordance with the approved national curriculum developed by the UK Foundation Programme Office,[1] but the topics listed are fairly broad and open to local interpretation. Specifically, there is no reference to the topics of medically unexplained symptoms (MUS) or clinical uncertainty within the curriculum headings, although 'communication in difficult circumstances' is likely to be of relevance to this subject.

MUS can be defined as symptoms not clearly linked to organic pathology, and include symptoms with no clear organic basis occurring within syndromes such as fibromyalgia, irritable bowel syndrome and

chronic fatigue.[2] MUS are common, accounting for up to 40%–50% of patients seen in primary care and up to half of patients in secondary care.[3–5] Consultations about such symptoms are often frustrating for both clinicians and patients due to the complexity and uncertainty around appropriate diagnosis and management,[6 7] and patients often express a fear of being dismissed or not believed by health professionals.[8] Unexplained symptoms are often linked to high levels of overinvestigation, referrals and unnecessary treatment, placing a significant financial strain on health services,[9 10] as well as the potential for iatrogenic harm to patients.[11] It is therefore essential that doctors become aware of appropriate management strategies and treatment options available for such patients early in their careers, as they are likely to represent a significant proportion of any doctor's clinical case load.

Recent studies have extensively reviewed the effectiveness of psychological and pharmacological treatments developed for patients with MUS,[12 13] although few studies have looked at the effectiveness of providing training for doctors working with these patients. Relevant training across undergraduate[14 15] and postgraduate[16] medical curricula within the UK is very limited, and attendance at teaching sessions is often not compulsory.

Training interventions for general practitioners (GPs) within postgraduate primary care in the UK[17] and various other European countries[18–20] have been developed, but are often brief, consisting of as little as a single session. The most thoroughly evaluated GP intervention in the UK, 'the reattribution technique', focuses on reattributing physical symptoms to psychosocial causes and was found to improve doctor–patient communication but had no positive impact on patient outcomes.[21] Randomised controlled trials conducted in Europe examining the effects of training interventions for GPs on patient outcomes have provided mixed results, with one showing improvement in several quality of life parameters, particularly bodily pain,[18] and another showing a significant reduction in patient visits at 6-month follow-up, but no improvement in patient outcomes.[20]

More recently, a 14-hour communication skills programme developed for qualified physicians in the Netherlands was found to improve doctor–patient communication across a variety of medical conditions, but patient outcomes were not assessed.[22] A review of recent studies looking at optimal management approaches for patients with MUS highlighted the importance of improving doctors' communication skills in this area and emphasised the need for clinicians to understand patients' expectations to be able to reduce their anxiety and improve overall satisfaction.[23] Delivering effective and empowering explanations for symptoms which are meaningful to both the doctor and patient is recommended, as well as providing appropriate levels of reassurance within the context of an empathic doctor–patient relationship.[23–25]

Education about MUS should ideally begin early in a clinician's training before management models and referral patterns are fully formed. The most opportune time may be during the early years immediately after qualification, as this is likely to be a time of significant clinical exposure to patients with MUS and junior doctors are often expected to make their own decisions regarding referrals and investigations for the first time. Effective training about the appropriate delivery of explanations for unexplained symptoms and suitable management approaches is needed,[16] although there is currently little consensus as to how this training should be delivered.

This study was part of a research project funded by the National Institute for Health Research School of Primary Care Research focusing on improving training about MUS for junior doctors undertaking the UK 2-year foundation programme (FY1/FY2). The first part involved qualitative in-depth interviews to examine junior doctors' experiences of managing patients with MUS and seek their recommendations for training and has been reported separately.[16] The second part is reported here, with the following three aims:

1. To assess to what extent teaching about MUS currently takes place within foundation training programmes (FTPs) in England, Wales and Northern Ireland and what this involves
2. To seek recommendations from FTP directors about the content, structure and length of future teaching sessions via a questionnaire survey
3. To hold an expert consultation workshop for professionals to synthesise these findings and formulate recommendations for a future educational intervention for junior doctors about managing patients with MUS

## METHODS

This study used a mixed-method design incorporating a national questionnaire survey and an expert consultation workshop.

### National questionnaire survey of FTPDs
#### Design

A web-based questionnaire entitled an 'Expert Consultation Exercise' (see online supplementary file) was designed comprising 14 questions. Its content was informed by the findings of the linked qualitative study[16] and an earlier survey of the provision of undergraduate teaching about MUS in UK medical schools.[15] Both open and closed questions asked for information on if, when and how teaching about MUS was delivered within the various postgraduate regions, as well as any perceived barriers to delivery, and suggestions for the content, structure and length of a proposed educational intervention on this topic.

#### Study population/setting

The questionnaire was administered by email link to 155 FTPDs at FTPs across England, Wales and Northern Ireland, whose names had been obtained via FTP websites. In the majority of cases contact email addresses

were not listed on the FTP websites, so email addresses were obtained via Google search.

## Data collection

Potential participants were sent an electronic request to complete the questionnaire using the online programme Survey Monkey (see http://www.surveymonkey.com). Two email reminders were sent at approximately one and two month intervals.

## Analysis

Responses to closed questions have been presented as descriptive statistics. Responses to open questions were analysed thematically to identify key themes emerging from the data. Three members of the research team (KY, MB and SH) independently identified key themes emerging from the dataset, then met and defined themes by consensus.

### Expert consultation workshop

#### Design

Programme directors who had completed the questionnaire were invited to attend a 3-hour expert consultation workshop, as were foundation level trainees involved in the linked qualitative interview study on the topic[16] and also medical educationalists who had attended a previous workshop on the topic of MUS at the 2011 Association for the Study of Medical Education (ASME) conference. The expert workshop aimed to discuss and synthesise the findings of the national questionnaire and the qualitative interview data from the linked study[16] to provide recommendations for the design and content of an educational intervention for junior doctors on this topic. Due to participant availability and time constraints, only one workshop date could be offered.

Workshop participants were given a summary of the questionnaire findings and the linked qualitative interview data. Participants were allocated into two groups, each with a trained facilitator (KY and MB) and encouraged to discuss the structure, content and length of future training for newly qualified doctors about MUS and any potential barriers to this. Discussion points were summarised and fed back to the main group.

#### Study population/setting

The workshop took place in Central London, UK. Thirteen people attended, including two programme directors, six GP educationalists, one medical sociologist, two FY2 junior doctors and two research associates representing four medical schools across the UK.

#### Data collection

Detailed notes were taken by two members of the research team (KY and SH).

#### Analysis

Following the workshop, a summary of the main discussion points was collated by two members of the research team (KY and SH) and distributed among workshop attendees. Attendees' comments and feedback were incorporated into a final agreed summary.

## RESULTS

### National questionnaire survey

Overall, responses were received from 53/155 (34.2%) of the FTPDs approached. Respondents included programme directors from 14 different specialties and represented 15 of the 19 foundation schools across England, Wales and Northern Ireland.

#### Current teaching on the topic of MUS

Details about current teaching taking place are documented in table 1. Nine of the 53 programme directors (17%) who responded to the questionnaire indicated that they were currently providing teaching about MUS within their foundation schools, although this only took place as a formal teaching session in six (11%) of these programmes. Of these, two-thirds were within Greater London.

#### Recommendations for training

Programme directors' recommendations regarding the proposed length and structure of an educational intervention are documented in table 2. On average they recommended 2.2 hours during FY1 and 2.7 hours during FY2, which would equate to one dedicated teaching session on the topic each year.

#### Potential barriers

Most programme directors who responded were very much in favour of delivering teaching about MUS within the foundation year programme, describing this

| Table 1 | Current teaching on the topic of MUS by foundation programmes within foundation schools |
|---------|---------|
| **Question** | **Response** |
| Is there teaching on MUS within FY1/2 training? | Yes (9/53), No (39/53), No response (5/53) |
| Is this a formal teaching session? | Yes (6/53), No (17/53), N/A (24/53), No response (10/53) |
| Is there any reference made to avoiding overinvestigation during MUS teaching or elsewhere? | Yes (21/53), No (23/53), No response (9/53) |
| Do these topics arise within case-based discussion or Balint-type groups? | Yes (22/53), No (19/53), No response (12/53) |

FY1/FY2, 1-year/2-year foundation programme; MUS, medically unexplained symptoms; NA, not applicable.

**Table 2** Questionnaire recommendations for future foundation level teaching about MUS

| Question | Response |
| --- | --- |
| What would you consider to be an ideal method of teaching? (Respondents may select more than one option) | Case-based group discussions (40/53)<br>GP/outpatient-based teaching (23/53)<br>Ward-based teaching (17/53)<br>Role-play with simulated patients (14/53)<br>Advanced consultation skills training (12/53)<br>Lectures/seminars (9/53)<br>Role-play with peers (8/53)<br>One-to-one supervision (6/53) |
| What would you consider to be the most feasible method of teaching? | Case-based group discussions (35/53)<br>Lectures (21/53)<br>GP/outpatient-based teaching (17/53)<br>Ward-based teaching (13/53)<br>Role-play with peers (7/53)<br>Role-play with simulated patients (6/53)<br>Advanced consultation skills training (5/53)<br>One-to-one supervision (4/53) |
| How many hours teaching on the topic of MUS would you recommend per year at: | FY1 level (mean: 2.24 hours; median: 2; range: 0–10)<br>FY2 level (mean: 2.67 hours; median: 2; range: 0–10.5) |

FY1/FY2, 1-year/2-year foundation programme; GP, general practitioner; MUS, medically unexplained symptoms.

as 'overdue' and an important topic in need of a formal teaching commitment.

> [This topic] should be part of required experiential teaching for Foundation trainees. (P52)

A few were more cautious, and concerned that focusing on MUS might lead to junior doctors overlooking diagnoses of organic disease. The need for training to be delivered by an experienced clinician and include a focused discussion about appropriate levels of investigation was emphasised.

> Juniors miss straightforward physical presentations that need investigations because their heads are filled with semi-psychiatric diagnosis in lieu of the need to investigate and treat the physical problems. (P46)

Some potential barriers to delivering the training included issues around time constraints and timetable space, a lack of resources and appropriate facilitators, scepticism from colleagues and junior doctor motivation to attend sessions.

> Time [as a barrier] - the topic requires unhurried exploration, especially including management strategies which often need to be individually 'tailored' (P52)

However, interestingly, several programme directors considered that there were 'no barriers' to such training and were keen to include this topic within the postgraduate educational curriculum as soon as possible.

### Expert consultation workshop

The topics discussed during the workshop, including recommendations for the length, content and structure of the proposed educational intervention, along with potential barriers to its delivery are outlined below in box 1.

### Length and content

Workshop attendees suggested the training should consist of two separate 2-hour teaching sessions. The first introductory session during the first year postqualification (FY1) should provide more factual content as a background to the topic, give definitions for the term MUS and raise awareness of both patient and clinician

---

**Box 1    Expert consultation workshop: key discussion points**

*Length*
Two separate 2-hour teaching sessions
*Content*
FY1: define MUS; raise awareness; emphasise clinical and economic implications
FY2: discuss clinical cases; provide examples of explanations for symptoms; address litigation fears and the impact of the potential negative attitudes of senior role models
*Structure*
Use of video vignettes to (1) illustrate positive and negative doctor–patient interactions and (2) show patients' lived experience of MUS
Case-based group discussions
Role-play
*Potential barriers and solutions*
Barriers: convincing colleagues of the topic's value and time constraints within educational curricula
Solution: emphasise prevalence of MUS to raise awareness among educationalists and senior clinicians and develop the relevant educational interventions
FY1/FY2, 1-year/2-year foundation programme; MUS, medically unexplained symptoms.

---

perspectives, with data illustrating the associated clinical and economic implications. The topic should then be revisited in the second year postqualification (FY2), with more emphasis on specific clinical cases or issues that participating doctors had experienced in dealing with patients with unexplained symptoms, with the opportunity to discuss examples of suitable physiological and psychological explanations for common symptom presentations. Fear of litigation was considered a potential significant source of anxiety for new doctors, and the importance of addressing any such concerns emphasised.

It was also thought important to raise awareness about the different attitudes, both helpful and unhelpful, which junior doctors might encounter from their senior colleagues concerning patients with MUS, with encouragement to reflect on the potential impact of these attitudes on their own views and resulting management choices. Due to its relevance to most specialties, it was suggested that some reference to MUS should also be made wherever appropriate throughout other foundation year educational sessions, although it was recognised this might be difficult to implement in practice.

### Structure
An innovative idea proposed by workshop attendees involved developing video vignettes to illustrate various doctor–patient interactions. For example, these could include positive and negative examples of role modelling when delivering explanations for common presentations of unexplained symptoms, and include other videos showing the lived experience of MUS from the patient perspective. Case-based group discussions, role-play and one-to-one supervision sessions focusing on issues around the identification and management of unexplained symptoms were also recommended. The preference was for face-to-face teaching to allow clinical case discussion, but developing an e-learning module incorporating including relevant video clips was also considered.

### Potential barriers to overcome
Convincing colleagues involved in the running of local foundation programmes about the topic's value in an already full curriculum was identified as a significant potential barrier to providing teaching to all newly qualified doctors. Emphasising the prevalence of MUS and raising awareness among educationalists and senior clinicians within relevant trusts was identified as an important step towards its inclusion in the postgraduate curriculum.

### DISCUSSION
Currently, teaching about MUS is not formally listed within the curriculum for newly qualified doctors in the UK. Very few questionnaire respondents reported any formal teaching on this topic within their foundation training programmes, and less than half of respondents reported it as being informally rather than systematically discussed in case-based discussions. These findings, together with a previous survey of medical undergraduate

teaching in this area,[15] and our linked study examining junior doctors' experiences of managing MUS,[16] indicate that teaching for both medical students and newly qualified doctors about this topic is currently very limited in the UK. This highlights an urgent need to adopt a more rigorous and systematic approach to education in this area.

Most programme directors recognised the importance of the topic and were in favour of integrating MUS into the postgraduate training curriculum. Some were highly enthusiastic and referred to such training as long overdue, while a few were more cautious and concerned about the potential for junior doctors to miss cases of organic disease.

Case-based group discussions were recommended as the most favourable teaching method by both questionnaire respondents and junior doctors in our linked qualitative study.[16] Practical ward-based or outpatient-based learning was also favoured by the programme directors who responded to the questionnaire, followed by role-play techniques involving simulated patients or peers. These findings correspond to a comprehensive 2011 review comparing 12 systematic reviews about the teaching of communication skills to qualified physicians, which reported that the most effective programmes often involve multiple training strategies.[26] Within the review, practice-based strategies which were longer and learner-centred were seen as most effective, and interactive methods including role-play, small group discussion and feedback were reported as having the most positive impact on learning. There is further evidence to suggest interactive methods such as case-based discussions are superior to bedside teaching and lectures,[27] and a more successful method of developing knowledge, influencing workplace practice[28] and stimulating interest.[29] Although role-play exercises have received mixed reviews from learners, this has been recognised as a useful way to hone skills and practice techniques in a safe setting.[16 22]

Another teaching method suggested by workshop participants was the use of videos to illustrate various doctor–patient interactions and demonstrate both positive and negative examples of role modelling. A recent study found that, after watching videos of patients describing disease-related symptoms, medical students developed better knowledge acquisition, a deeper understanding of the problem and showed increased interest in the patient.[30] Using several techniques, such as incorporating case-based discussions with video work, facilitates learning more than the use of a single technique[31 32] and can lead to improved clinical outcomes,[33] as information is reinforced through the use of different techniques which appeal to a variety of learning styles.

In the present study, workshop participants suggested providing a 2-hour teaching session during the first year following qualification and then revisiting the topic during the second year, as well as referring to MUS where relevant throughout the curriculum. This is an approach supported by research showing that multiple exposures

to the same subject matter over time can lead to greater knowledge gain and facilitate more positive attitudes towards learning.[31 32]

In light of the results of this study and the wider literature, factors to consider for an educational intervention for newly qualified doctors include raising awareness about the topic, assisting doctors in the recognition of patients with MUS and providing information about effective management strategies appropriate for the level of contact. This would focus mainly on accurately identifying the problem and the patient's concerns, giving effective explanations for symptoms which make sense to both the patient and the practitioner, providing appropriate reassurance and demonstrating empathy as well as avoiding unnecessary investigations and referrals.[23–25] Highlighting that patients with MUS appear to seek emotional support more than other patients is also important,[6] as this may contribute to the difficulties some of the junior doctors experienced when working with these patients in our linked qualitative study(16). A number of these current deficiencies in training were also highlighted by junior doctors in the linked study[16] as juniors spoke about feeling stuck and unsure about how to construct and deliver suitable explanations, and feared patients' reactions to negative test results or unclear diagnoses. Including teaching about various explanatory models of MUS, such as somatosensory amplification theory, immune system sensitisation theory and various cognitive theories,[34] could also be useful to encourage doctors to think about providing explanations for MUS within a biopsychosocial context.[16] In light of recent Cochrane reviews examining effective psychological interventions for patients experiencing MUS (eg, cognitive behavioural therapy or psychodynamic therapies),[12 13] raising junior doctors' awareness about possible treatments and referral options would be an important component of training[35].

It is important to highlight that, although specific management techniques have been recommended in our paper and in the literature, there is currently only clear evidence for their effectiveness in improving clinician skills when communicating with patients with MUS[22] and reducing investigations and healthcare costs.[20] An educational intervention focusing on these areas is likely to produce tangible benefits in terms of reduced frustration for both patients and clinicians, increased patient satisfaction and reduced costs. The evidence for a direct impact on clinical outcomes such as improved mood, functioning or quality of life is still lacking and any formal evaluation of a new educational intervention would need to carefully assess these factors. The combination of videos and case-based learning which we are suggesting is a novel approach and might prove more effective at impacting on clinical outcomes given the evidence for such a combined approach.[31–33]

Studies have highlighted the impact of the negative views of some senior role models on juniors' attitudes and management choices,[14 16] drawing attention to this wider issue and the need to bring the effective management of MUS to the attention of doctors of all levels if any training interventions are to be successfully implemented.

## Strengths and limitations

To the best of our knowledge, there has been no previous research into the provision of early postgraduate teaching on the topic of MUS across foundation schools internationally or the views of programme directors regarding future education in this topic. Only around a third of programme directors approached participated in the survey, but participants were forthcoming with both their positive views and any reservations, and 15 out of 19 of the foundation schools across England, Wales and Northern Ireland were represented in the responses. It is possible that email addresses retrieved from the internet may be outdated, meaning a number of programme directors may not have received the email. As the topic of MUS does not currently feature on the core foundation school curriculum and the actual scale of provision of teaching on the topic of MUS within the remaining foundation schools remains unknown, strong conclusions regarding the rates of teaching nationally cannot be confidently drawn from these data. It may be that those who responded to the questionnaire held more positive views towards MUS and its importance within the curriculum.

## Implications for future research

Future research should focus on developing an educational programme aimed at newly qualified doctors which could become part of the national curriculum, and evaluating this in terms of its impact on patient and doctor satisfaction. It would also be important to establish whether there is a positive impact in relation to reduced cost of investigations, repeated patient attendances and patient outcomes in terms of physical and mental health.

**Contributor** MB, SN, KRW, JR and AW developed the study protocol. All authors attended the workshop. KY collected the data. KY, MB, SN, KRW, KL and SH analysed the data and interpreted the results. KY and SH drafted the report. All authors have read and approved the manuscript.

**Funding** This work was supported by the National School of Primary Care Research (NSPCR) within the National Institute for Health Research (NIHR). Award Number: 157946.

**Disclaimer** The views expressed are those of the author(s) and not necessarily those of the NHS, the NIHR or the Department of Health. The funder has had no role in the study design, collection, analysis and interpretation of data, writing of the manuscript or decision to submit for publication.

**Competing interests** MB, KW, JR and AW provide lectures on the topic of MUS for fourth-year and fifth-year medical students and foundation year doctors in the Department of Primary Care and Population Health, UCL.

**Ethics approval** UCL Research Ethics Committee.

**Provenance and peer review** Not commissioned; externally peer reviewed.

**Data sharing statement** An anonymised summary of the workshop discussion points and participant comments is available upon request. An anonymised summary of all of the quantitative and qualitative questionnaire responses received is also available upon request from the corresponding author.

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
