## [Reviewer comments · BMJ Open]

ARTICLE DETAILS

TITLE (PROVISIONAL)	Improving teaching about medically unexplained symptoms for newly-qualified doctors in the UK: findings from a questionnaire survey and expert workshop
AUTHORS	Yon, Katherine; Habermann, Stephanie; Rosenthal, Joe; Walters, Kate; Nettleton, Sarah; Warner, Alex; Lamahewa, Kethakie; Buszewicz, Marta

VERSION 1 - REVIEW

REVIEWER	Helen Salisbury University of Oxford UK
REVIEW RETURNED	08-Nov-2016

GENERAL COMMENTS	I could not access the appendix so I could not view the questionnaire as I would have liked to. This is an important topic and clearly there is some consensus that it is a gap in the education of junior doctors. There is a slight concern that the minority of PDs who did reply to the survey were the enthusiasts and there may be less consensus that this study suggests. The paper is clearly and simply written and easy to follow. The section on the possible form of teaching was particularly interesting. There remain many unanswered questions about the nature of medically unexplained symptoms and specifically how they should be explained to patients. However the advantage of the case based discussion and role play styles of teaching presented is that it enables young doctors to explore these uncertainties. I think this paper contributes usefully to the field.
---

REVIEWER	Christopher Burton University of Aberdeen UK
REVIEW RETURNED	14-Nov-2016

GENERAL COMMENTS	This is a clearly written piece describing the results of a nonetheless potentially important consultation exercise. Design strengths include wide sampling (with moderate response) and inclusion of an interactive workshop (although there was only one and attendance was modest). Background is reasonable and cites appropriate systematic reviews (although then goes on to cherry pick findings from some included studies in a rather selective way). However the pervasive argument that it is a good thing to train junior doctors about MUS, needs to be tempered with explicit statements that we don't yet know enough
--

	about what works for MUS in routine consultations and so should acknowledge this. One reading of the Rosendal review - and observational data from studies such as those included in that review – is that while doctors can be taught to deliver better communication it may not affect patient outcomes. Methods are appropriate and do not overstate the findings. Results are reasonable – although while the authors describe reporting “structure and content” it seems to be more about structure and delivery rather than the actual content. It is difficult to see what would be the core learning outcomes here and how the authors derived these from data. There is no attempt to address issues raised in reactions to undergraduate training such as the “these are just symptoms, where’s the science?” which the authors cite. Overall, if I was designing a foundation educational programme, this paper would leave me thinking that my peers probably thought I should do something about MUS, and how they might deliver it, but I wouldn’t really be clear what I should deliver. So it is useful, I wonder if it could be more useful while staying grounded in the data?
--	--

VERSION 1 – AUTHOR RESPONSE

Reviewer: 1

Reviewer Name: Helen Salisbury

Institution and Country: University of Oxford, UK

Competing Interests: None declared

- 1. I could not access the appendix so I could not view the questionnaire as I would have liked to. This is an important topic and clearly there is some consensus that it is a gap in the education of junior doctors. There is a slight concern that the minority of PDs who did reply to the survey were the enthusiasts and there may be less consensus that this study suggests. The paper is clearly and simply written and easy to follow. The section on the possible form of teaching was particularly interesting. There remain many unanswered questions about the nature of medically unexplained symptoms and specifically how they should be explained to patients. However the advantage of the case based discussion and role play styles of teaching presented is that it enables young doctors to explore these uncertainties. I think this paper contributes usefully to the field.**

We are delighted that the reviewer has acknowledged the importance of this topic and the paper’s useful contribution to the field. We would like to thank her for the comments on the content of the paper and its presentation. As has been pointed out, we recognise that those responding to the survey might have been more enthusiastic about or invested in the topic of MUS. We have addressed this point within the ‘strengths and limitations’ section on page 18, paragraph 2:

As the topic of MUS does not currently feature on the core Foundation School curriculum and the actual scale of provision of teaching on the topic of MUS within the remaining Foundation Schools remains unknown, strong conclusions regarding the rates of teaching nationally cannot be confidently drawn from this data. It may be that those who responded to the questionnaire held more positive views towards MUS and its importance within the curriculum.

Reviewer: 2

Reviewer Name: Christopher Burton

Institution and Country: University of Aberdeen UK

Competing Interests: None

1. This is a clearly written piece describing the results of a nonetheless potentially important consultation exercise. Design strengths include wide sampling (with moderate response) and inclusion of an interactive workshop (although there was only one and attendance was modest). Background is reasonable and cites appropriate systematic reviews (although then goes on to cherry pick findings from some included studies in a rather selective way). However the pervasive argument that it is a good thing to train junior doctors about MUS, needs to be tempered with explicit statements that we don't yet know enough about what works for MUS in routine consultations and so should acknowledge this. One reading of the Rosendal review – and observational data from studies such as those included in that review – is that while doctors can be taught to deliver better communication it may not affect patient outcomes.

We would like to thank the reviewer for this positive feedback about the paper. We agree that it is important to emphasise that we do not yet have enough information about the effects of the management strategies being suggested on patient clinical outcomes. In light of this, we have amended the paper on page 17, paragraph 2 to clarify what is currently known about the impact of such management strategies on patient and clinician communication and health-care costs, but acknowledge that no educational intervention to date has had a direct impact on clinical outcomes and that these outcomes would need to be assessed with any new educational intervention:

It is important to highlight that although specific management techniques have been recommended in our paper and in the literature, there is currently only clear evidence for their effectiveness in improving clinician skills when communicating with patients with MUS[22] and reducing investigations and health-care costs.[20] An educational intervention focusing on these areas is likely to produce tangible benefits in terms of reduced frustration for both patients and clinicians, increased patient satisfaction and reduced costs. The evidence for a direct impact on clinical outcomes such as improved mood, functioning or quality of life is still lacking and any formal evaluation of a new educational intervention would need to carefully assess these factors. The combination of videos and case-based learning which we are suggesting is a novel approach and might prove more effective at impacting on clinical outcomes given the evidence for such a combined approach.[31-33]

2. Methods are appropriate and do not overstate the findings. Results are reasonable – although while the authors describe reporting “structure and content” it seems to be more about structure and delivery rather than the actual content. It is difficult to see what would be the core learning outcomes here and how the authors derived these from data. There is no attempt to address issues raised in reactions to undergraduate training such as the “these are just symptoms, where's the science?” which the authors cite. Overall, if I was designing a foundation educational programme, this paper would leave me thinking that my peers probably thought I should do something about MUS, and how they might deliver it, but I wouldn't really be clear what I should deliver. So it is useful, I wonder if it could be more useful while staying grounded in the data?

We acknowledge the reviewer's comment regarding our recommendations for training content. We have drawn on the findings of other studies, and recommendations resulting from this study, to highlight factors which would be useful to include in a training programme (see page 16, paragraph 4; page 17, paragraph 1).

*In light of the results of this study and the wider literature, factors to consider for an educational intervention for newly qualified doctors include raising awareness about the topic, assisting doctors in the recognition of patients with MUS, and providing information about effective management strategies appropriate for the level of contact. **This would focus mainly on accurately identifying the problem and the patient's concerns, giving effective explanations for symptoms which make sense to both the patient and the practitioner, providing appropriate reassurance and demonstrating empathy, as well as avoiding unnecessary investigations and referrals.**[23-25]*

In response to the query about the lack of discussion of any scientific theories, we have also added in a sentence to suggest the inclusion of teaching about scientific theories for MUS on page 17, paragraph 1:

*Highlighting that patients with MUS appear to seek emotional support more than other patients is also important,[6] as this may contribute to the difficulties which some of the junior doctors experienced when working with these patients. A number of these current deficiencies in training were highlighted by junior doctors in our linked qualitative study,[16] as juniors spoke about feeling stuck and unsure about how to construct and deliver suitable explanations, and feared patients' reactions to negative test results or unclear diagnoses. **Including teaching about various explanatory models of MUS, such as somatosensory amplification theory, immune system sensitisation theory and various cognitive theories [34], could be useful to encourage doctors to think about providing explanations for MUS within a biopsychosocial context [16].** In light of recent Cochrane reviews examining effective psychological interventions for patients experiencing MUS (e.g. cognitive behavioural therapy or psychodynamic therapies),[12-13] raising junior doctors' awareness about possible treatments and referral options would also be an important component of training.*

34. Ravenzwaaij J van, olde Hartman TC, Ravesteijn H van, Eveleigh R, Rijswijk E van, Lucassen PLBJ. Explanatory models of medically unexplained symptoms: a qualitative analysis of the literature. Ment Health Fam Med 2010; 7:223-31.

I look forward to hearing from you.

Yours sincerely,

Katherine Yon (on behalf of all authors)

VERSION 2 – REVIEW

REVIEWER	Christopher Burton University of Sheffield
REVIEW RETURNED	01-Mar-2017
GENERAL COMMENTS	Revisions have addressed issues satisfactorily